# Design and Simulation of High-Temperature Micro-Hotplate for Synthesis of Graphene Using uCVD Method

**DOI:** 10.3390/mi15040445

**Published:** 2024-03-27

**Authors:** Lvqing Bi, Bo Hu, Dehui Lin, Siqian Xie, Haiyan Yang, Donghui Guo

**Affiliations:** 1School of Electronic Science and Engineering, Xiamen University, Xiamen 361005, China; bilvqing108@ylu.edu.cn (L.B.);; 2Guangxi Colleges and Universities Key Laboratory of Complex System Optimization and Big Data Processing, Yulin Normal University, Yulin 537000, China; 3Guangxi Center for Applied Mathematics, Yulin Normal University, Yulin 537000, China

**Keywords:** micro-hotplate, microchemical vapor deposition (uCVD), temperature uniformity, suspended multi-cantilever, selective doping process

## Abstract

The uCVD (microchemical vapor deposition) graphene growth system is an improved CVD system that is suitable for scientific research and experimental needs, and it is characterized by its rapid, convenient, compact, and low-cost features. The micro-hotplate based on an SOI wafer is the core component of this system. To meet the requirements of the uCVD system for the micro-hotplate, we propose a suspended multi-cantilever heating platform composed of a heating chip, cantilevers, and bracket. In this article, using heat transfer theory and thermoelectric simulation, we demonstrate that the silicon resistivity, current input cross-sectional size, and the convective heat transfer coefficient have a huge impact on the performance of the micro-heating platform. Therefore, in the proposed solution, we adopt a selective doping process to achieve a differentiated configuration of silicon resistivity in the cantilevers and heating chip, ensuring that the heating chip meets the requirements for graphene synthesis while allowing the cantilevers to withstand high currents without damage. Additionally, by adding brackets, the surfaces of the micro-hotplate have the same convective heat transfer environment, reducing the surface temperature difference, and improving the cooling rate. The simulation results indicate that the temperature on the micro-hotplate surface can reach 1050.8 °C, and the maximum temperature difference at different points on the surface is less than 2 °C, which effectively meets the requirements for the CVD growth of graphene using Cu as the catalyst.

## 1. Introduction

Graphene is a two-dimensional carbon allotrope material with a hexagonal honeycomb lattice structure and single-atom thickness, as shown in Figure 1 [1]. Since Geim and Novoselov obtained graphene through mechanical exfoliation in 2004 [2], graphene has attracted extensive attention and research due to its outstanding electrical, thermal, optical, and mechanical properties, and it has potential applications in microelectronics, biosensing, medicine, renewable energy, RF systems, and other fields [3,4,5,6,7,8]. Moreover, graphene’s ultra-high electron mobility and stable physical properties make it an ideal nanomaterial for replacing silicon in the future [9,10,11]. However, before this, high-quality graphene must be prepared under controlled conditions.

Nowadays, various methods for graphene preparation have been disclosed and confirmed, which can be classified into two types: top–down and bottom–up. The top–down methods include the following nine types: mechanical exfoliation, graphite intercalation, carbon nanotube slicing, thermal decomposition, reduction of graphene oxide (GO), electrochemical exfoliation, ultrasonic separation, ball milling, and laser irradiation. On the other hand, the bottom–up methods consist of the following four types: metal–carbon solid growth, SiC epitaxial growth, dry ice method, and deposition [12]. Among them, chemical vapor deposition (CVD) in the bottom–up deposition methods has become the most popular approach for graphene synthesis due to its potential for large-scale production, well-established industrial infrastructure, and large-area preparation [13,14], especially the CVD synthesis of graphene using Cu and Ni as catalysts [15,16,17,18].

The concept of the microchemical vapor deposition (uCVD) preparation system was first proposed by researchers from Berkeley University in 2009, which is an improved method of chemical vapor deposition (CVD) [19]. In addition to the various advantages of CVD, it avoids the drawbacks of long reaction time, low growth efficiency, long experimental cycles, and high experimental costs [20,21]. It also has characteristics of programmability, miniaturization, and rapid preparation. In the uCVD method, the suspended multi-cantilever micro-hotplate used for heating the catalyst and growing graphene on the surface is the core component of the preparation system, and it is also the most crucial part affecting the quality of graphene growth. However, in this uCVD system, the cantilever, due to its slender structure, has a much higher temperature than the central heating block, making it not only prone to fracture but also damaging the temperature uniformity of the entire micro-hotplate, affecting the quality of graphene synthesis. Therefore, we have set out to design a new micro-hotplate in this paper that effectively meets the requirements of the uCVD system.

The electrical and mechanical stability of the device structure should be considered as well due to the drastic temperature and surrounding airflow changes that occur in micrometer-scale devices in a short period. In this paper, a thermal–electric model of ANSYS Workbench is used to analyze the impact of structure and material resistivity on the heating performance of the micro-hotplate. Computer simulation is widely used in the design of MESM devices [22]. It can not only provide verification for theoretical design but also provide convenience for optimized design and improve design efficiency. The thermal–electric simulation of ANSYS Workbench has flexible structural design functions and convenient material parameters and environmental convection parameters settings. Setting the resistivity of the heating chip and cantilever Si to 0.2Ω·cm and 0.04Ω·cm, respectively, the device length and width are both 5 mm; the convective heat transfer coefficientunder is 20 W/(m^2^·k); a constant current (1 A) is inputted, the temperature on the heating chip surface can reach 1050.8 °C, and the maximum temperature difference at different points on the surface is less than 2 °C, which effectively meets the requirements for the CVD growth of graphene using Cu as the catalyst, shortens the preparation time, and improves the consistency of the produced graphene.

## 2. The uCVD Graphene Synthesis System

The uCVD graphene preparation system is developed by referring to existing CVD systems [19,23] and optimizing the design. Although the basic principles of preparation are the same, the system described in this paper has the characteristics of being fast, convenient, compact, and low cost, making it more suitable for the needs of research institutes, universities, and factory laboratories. The system diagram is shown in Figure 2.

In the figure above, the micro-hotplate, as the core component of the system, is required to complete various growth processes, including high-temperature annealing (in an H_2_ environment at 1000 °C for 20 min), constant temperature growth maintenance (in a CH_4_ and H_2_ mixed environment at 1000 °C for 5 min), and rapid cooling (controlled within 1 min from 1000 °C to room temperature). Additionally, the temperature difference on the heating chip surface should be minimized to synthesize high-quality graphene. Therefore, the heating chip should possess characteristics such as miniaturization, high electrical resistivity, and low thermal expansion.

## 3. Analytical Models

### Electrothermal Model of the Micro-Hotplate in uCVD System

As shown in Figure 3, the designed micro-hotplate is based on an SOI wafer with a Si substrate thickness of 500 μm, a SiO_2_ dielectric layer thickness of 1 μm, and a surface copper layer thickness of 30 μm. Place the micro-hotplate into a sealed cavity and heat it with electricity. According to Joule’s law, the energy generated is
(1)$$Q=I^{2}Rt$$
where *I* is the input current, *t* is the heating duration, *R* is the impedance of the substrate, and $R=\rho \frac{L}{S}$ (ρ is the resistivity of the boron-doped Si substrate, *L* is the micro-hotplate length, and *S* is the effective cross-sectional area of the current input surface). According to the laws of thermodynamics, in the early stage of vacuum annealing (temperature rise stage), the thermal energy generated by electrification is not only absorbed by the micro-hotplate to increase internal energy (temperature rise) but also lost due to thermal radiation and convection, so
(2)$$Q=Q_{A}+Q_{R}+Q_{C}$$
where $Q_{A}$ is the added internal energy, and $Q_{R}$ and $Q_{C}$ represent the energy losses due to thermal radiation and thermal convection, respectively. Due to the small size of the micro-hotplate, the thermal conductivity of the materials used is high, and the layers are in close contact; thus, it can be approximated that the overall temperature of the micro-hotplate is uniformly distributed. Therefore, the increase in internal energy can be calculated using the following equation: (3)$$Q_{A}=\sum_{i=1}^{3}c_{i}m_{i}\Delta T\left(t\right)$$
where $c_{i}$ and $m_{i}$ are the specific heat capacity and mass of each layer material, respectively; ΔT(t) is the temperature difference before and after heating. The thermal radiation and thermal convection can be described by the following equations: (4)$$Q_{R}=\int_{0}^{t}\sum_{i=1}^{3}\varepsilon_{i}A_{i}\sigma \left(T^{4}\left(\tau \right)-T_{S}^{4}\right)d_{\tau }$$
(5)$$Q_{C}=\int_{0}^{t}\sum_{i=1}^{3}hA_{i}\left(T\left(\tau \right)-T_{\infty }\right)d_{\tau }$$
where σ is the Boltzmann constant, $\varepsilon_{i}$ and $A_{i}$ are the emissivity and surface area of the *i*th layer material, *h* is the convective heat transfer coefficient, and $T_{S}$ and $T_{\infty }$ are the temperatures of the inner wall and gas inside the cavity, respectively.

## 4. Simulation and Optimization

### 4.1. Effect of Silicon Resistivity on Heating

When the structure of the micro-hotplate and input current remains unchanged, a numerical iterative method can be used to approximately solute the heating curve. Taking the time interval as Δt, Equation (Equation 2) can be approximated as
(6)$$I^{2}R\cdot \Delta t=\sum_{i=1}^{3}\sum_{n=1}^{\infty }\left(c_{i}m_{i}\left(T_{n}-T_{n-1}\right)+\varepsilon_{i}A_{i}\sigma \left(T_{n-1}^{4}-T_{S}^{4}\right)\Delta t+hA_{i}\left(T_{n-1}-T_{\infty }\right)\Delta t\right)$$

From this, it can be deduced that the heating temperature $T_{n}$ is
(7)$$T_{n}=\frac{I^{2}\rho \frac{L}{S}\cdot \Delta t-\sum_{i=1}^{3}\varepsilon_{i}A_{i}\sigma \left(T_{n-1}^{4}-T_{S}^{4}\right)\Delta t-\sum_{i=1}^{3}hA_{i}\left(T_{n-1}-T_{\infty }\right)\Delta t}{\sum_{i=1}^{3}c_{i}m_{i}}+T_{n-1}$$

In the above equations, n=1,2,3,⋯, $T_{n-1}$ and $T_{n}$ represent the initial temperature and final temperature of the *n*th heating period, $T_{0}=T_{S}=T_{\infty }=25$ °C. The simulation curves of heating temperature under different resistivity are shown in Figure 4. When the simulation conditions are within certain parameters (input current I= 1.1 A), the length and width of the micro-hotplate are both 5 mm, the thickness of the Si, SiO_2_, and Cu layers are H1 = 300 μm, H2 = 1 μm, and H3 = 25 μm, and the material emissivity is referred to in references [24,25]. The simulation curves of heating temperature under different resistivities of silicon layers are shown in Figure 4.

As shown in the above figure, it can be observed that the higher the resistivity of the conductive silicon layer, the steeper the temperature curve rises, and the shorter the time required to heat up to the predetermined temperature. When the resistivity is too high, the heating temperature greatly exceeds the melting point of Cu, resulting in damage to the micro-hotplate while also increasing the difficulty of control due to the short heating time. However, if the resistivity is too low, it may not be possible to heat up to the expected temperature value. Therefore, combining the structural parameters and obtaining an appropriate resistivity through rational doping is a key point in the design of a micro-hotplate.

### 4.2. Effect of the Structural Parameters of Micro-Hotplate on Heating Performance

Heating plates based on the SOI structure often have fixed layer thicknesses, so the adjustment of structural dimensions mainly focuses on the length and width. When other parameters remain unchanged, the relationship between heating temperature and micro-hotplate length and width dimensions is shown in Figure 5.

From Figure 5, it can be seen that as the micro-hotplate length increases, the heating speed does not significantly increase, but the maximum achievable heating temperature of the micro-hotplate increases due to the increase in the resistance of the conductive Si layer. When the micro-hotplate length is increased to a certain value (greater than 5 mm), the increase in the maximum achievable heating temperature is no longer significant. Therefore, the length of the micro-hotplate can be appropriately increased as needed to synthesize longer graphene without worrying about damage, since its surface temperature exceeds the melting point of Cu. Compared to the length of the hotplate, the width of the current input cross-section has a more pronounced effect on heating. Reducing the width rapidly increases the resistance of the conductive Si layer, resulting in a rapid increase in heating speed and a sharp increase in the maximum achievable heating temperature, which may cause the surface temperature of the micro-hotplate to exceed the melting point of Cu.

### 4.3. Effect of the Convective Heat Transfer Coefficient on Heating Performance

When preparing graphene using the uCVD system, it is necessary to force the carbon source gas (reaction gas) and carrier gas into the reaction chamber and control the flow rate. Similar to regular CVD systems, laminar flow is usually preferred for gas flow in uCVD. The type and flow rate of gases vary in different process stages with corresponding convective heat transfer coefficient ranging from 10 to 100 W/(m^2^·k). Therefore, we discussed the effect of gas flow rate on the heating performance of the micro-hotplate within this scope, as shown in Figure 6.

It can be seen from Figure 6 that during the heating stage, the effects of different convective heat transfer coefficients (corresponding to the gas flow rates) on heating performance are approximately the same. However, in the constant temperature stage, the gas flow rate is inversely related to the maximum achievable heating temperature. That is, a higher maximum achievable heating temperature can be obtained at a small flow rate (corresponding to a small convective heat transfer coefficient), and conversely, a lower maximum achievable heating temperature may even make it impossible to meet the synthesis process requirements.

### 4.4. Structural Optimization

To verify the temperature characteristics of the SOI-based micro-hotplate mentioned above, we conducted thermoelectric simulation using the same size parameters (the resistivity of the conductive Si layer is 0.2 Ω·cm) on the commercial software Ansys 2021.R1. The results are shown in Figure 7.

As can be seen from the above figure, the temperature of electric heating is about 1000 °C, which is highly close to the results demonstrated in the previous section. In addition, the temperature distribution on the surface of the top Cu layer is uniform. Although the temperature in the center is higher and the temperature around it is lower, the temperature difference across the entire surface is only about 2 °C, which improves the consistency of the synthesized graphene.

To facilitate input current, improve the uniformity of the micro-hotplate surface temperature, and speed up the heating speed, the improved suspended micro-hotplate is shown in Figure 8.

As shown in the above figure, a bottom bracket is constructed using Si and SiO_2_, allowing the heating platform to be suspended on it. The suspended heating structure enables each surface to have the same convective heat transfer environment, improving the temperature consistency of the micro-hotplate and accelerating the cooling rate during the cooling stage. In addition, to avoid the frequent occurrence of cantilever fractures caused by high temperatures in the experiment, we adopted a selective doping process. After selecting a boron-doped N-type silicon layer with a resistivity of 0.2 Ω·cm as the basic power layer, a heavily doped thin layer with a resistivity of 0.004 Ω·cm was formed on the surface of the cantilever through photolithography and diffusion processes. This allows the cantilever to not only pass a larger current but also concentrate the heat generated by the micro-hotplate mainly on the synthetic block in the middle. The thermal–electric simulation results of the improved micro-hotplate are shown in Figure 9.

From the simulation results in the above figure, it can be seen that the suspended multi-cantilever heating structure using selective doping technology not only improves the temperature distribution of the top layer, maintains better temperature consistency, and reduces the temperature difference of the synthesis surface, it also lowers the temperature of the cantilever, making the heating platform more convenient, stable, and reliable, meeting the strict requirements of the graphene growth process.

## 5. Fabrication Process of the Micro-Hotplate

A simplified fabrication process flow for the suspended multi-cantilever micro-hotplate is illustrated in Figure 10. In this fabrication process, an SOI wafer is used as a temporary carrier with a thickness of 500 μm and a Si resistivity of 0.2 Ω·cm (Figure 10a). Subsequently, using the prepared mask, the photolithography process is used to cover the required mask pattern on the SOI, and the DRIE process is used to etch away the excess silicon in the top and bottom substrates (Figure 10b). Then, SiO_2_ in the cantilever region was etched with HF (Figure 10c). Next, a 1 μm thick SiO_2_ layer was deposited on the surface of the device using the LPCVD system (Figure 10d). Afterwards, a 25 μm thick Cu layer was grown on the surface of the device using an electron beam evaporation process (Figure 10e). After that, the same process technology as Figure 10b,c is used to etch away the Cu and SiO_2_ on the cantilever (Figure 10f). Finally, using the doping process, boron ions are doped on the surface of the cantilever (with a doping depth of 50 μm), resulting in a Si resistivity of 0.004 Ω·cm on the surface of the cantilever (Figure 10g).

## 6. Conclusions

In this paper, a suspended multi-cantilever heating platform composed of a heating chip, cantilevers, and bracket is proposed. It not only meets the requirements of the uCVD graphene synthesis system; the fabrication process is also compatible with a standard CMOS SOI process, which can meet the needs of rapid preparation of graphene in scientific research and experiments.

We use heat transfer theory to study the heating principle of the basic SOI micro-hotplate and explain the influence of the resistivity of the Si layer, the device structure size, and the convective heat transfer coefficient on the heating performance. In addition, the 3D finite element simulation using ANSYS shows that the micro-hotplate has good temperature consistency. To facilitate current input and improve temperature distribution consistency and heating efficiency, we propose a suspended multi-cantilever improved heating platform structure using selective doping technology, which achieves better temperature consistency, shortens heating and cooling time, and well meets the requirements of the uCVD graphene preparation system. The simulation results show that with a constant current input of 1 A, the surface temperature of the micro-hotplate can reach 1050.8 °C, and the temperature difference is only 2 °C.

In addition, we also provide a simplified fabrication process flow for the micro-hotplate, which is compatible with the standard CMOS SOI process, providing a reference for future industrialization.

## Figures and Tables

**Figure 1 micromachines-15-00445-f001:**
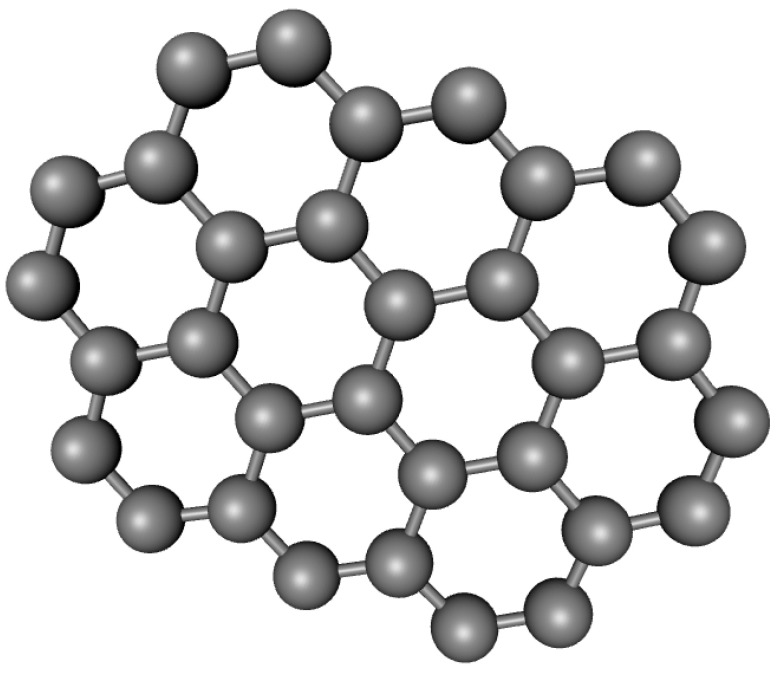
Graphene with two-dimensional structure.

**Figure 2 micromachines-15-00445-f002:**
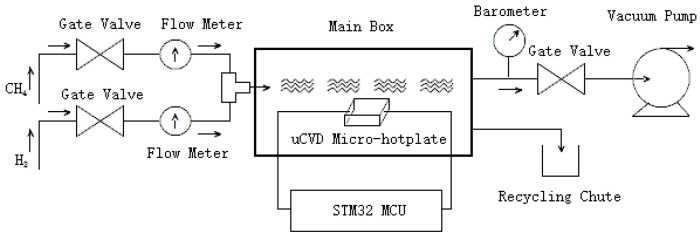
The block diagram of the uCVD system.

**Figure 3 micromachines-15-00445-f003:**
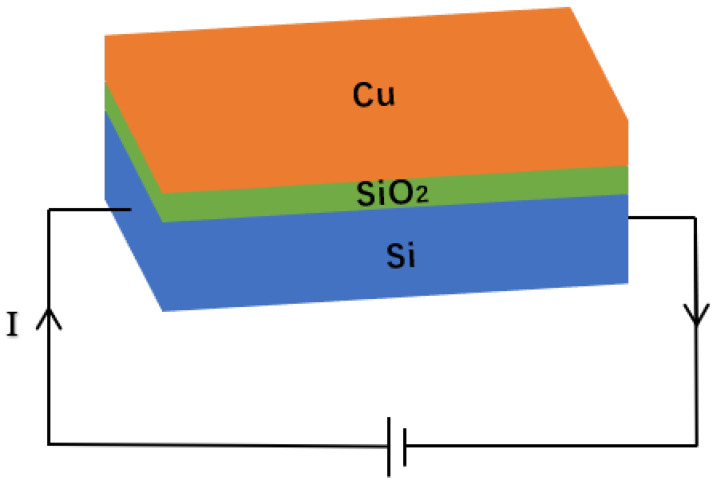
The basic structure of the micro-hotplate.

**Figure 4 micromachines-15-00445-f004:**
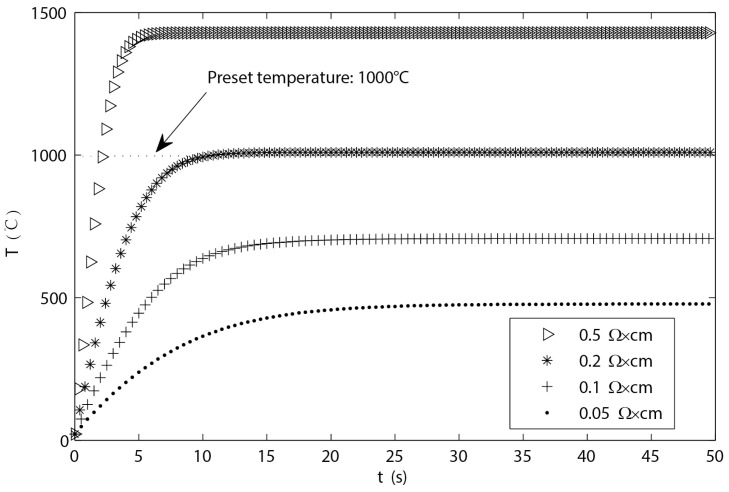
The temperature curves for heating at different resistivities of silicon layers.

**Figure 5 micromachines-15-00445-f005:**
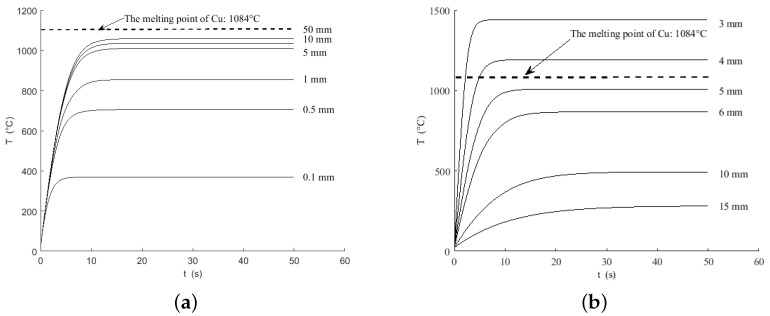
Heating temperature curves for different micro-hotplate sizes: (**a**) difference length; (**b**) difference width.

**Figure 6 micromachines-15-00445-f006:**
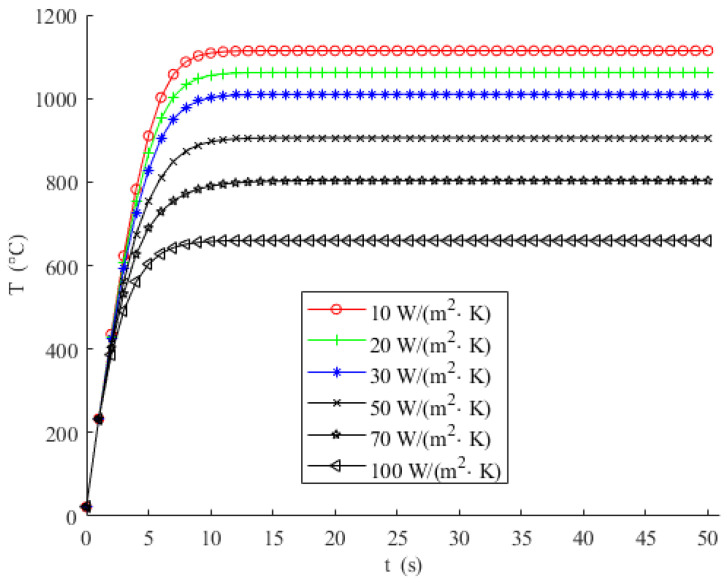
Heating temperature curves with different convective heat transfer coefficient.

**Figure 7 micromachines-15-00445-f007:**
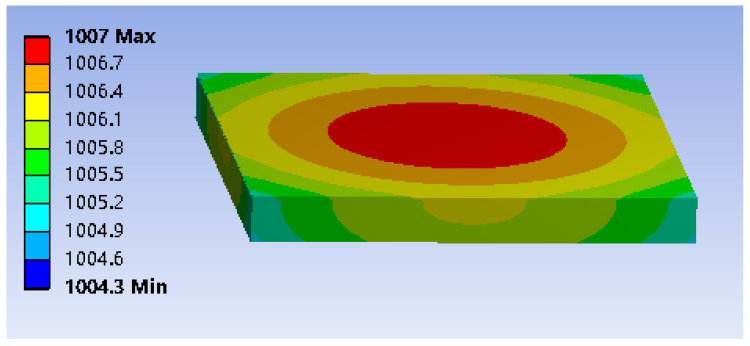
Thermoelectric simulation results of the micro-hotplate.

**Figure 8 micromachines-15-00445-f008:**
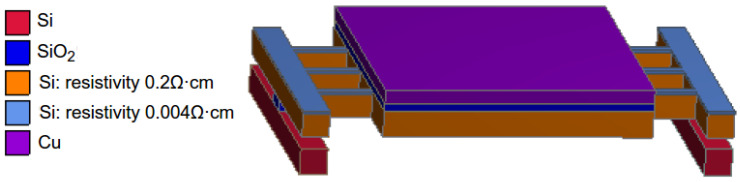
Suspended multi-cantilever micro-hotplate.

**Figure 9 micromachines-15-00445-f009:**
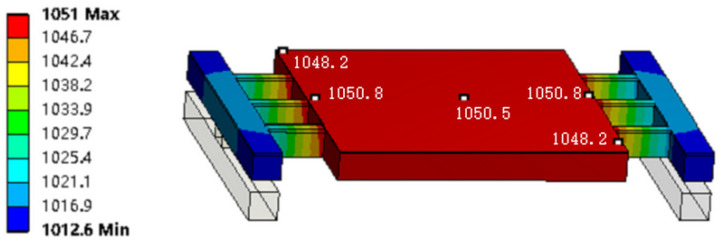
Thermal–electric simulation of suspended multi-cantilever micro-hotplate.

**Figure 10 micromachines-15-00445-f010:**
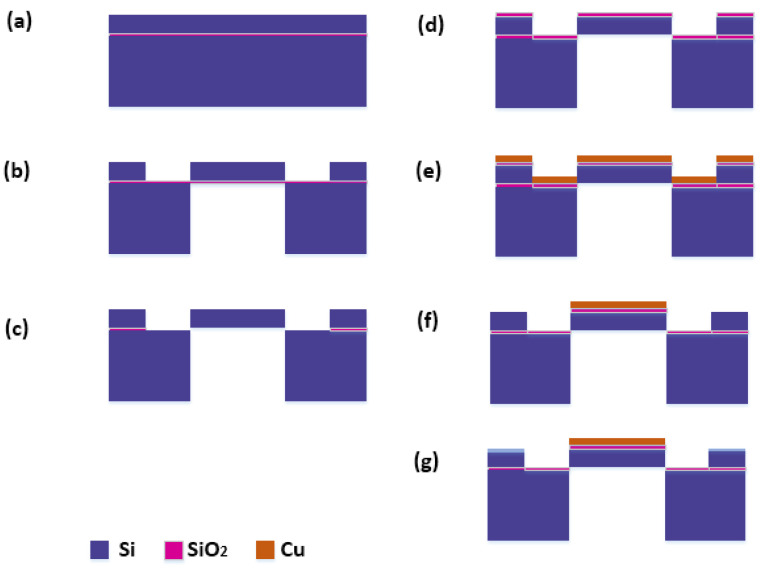
Simplified fabrication process flow for suspended multi-cantilever micro-hotplate: (**a**) the SOI wafer carrier; (**b**) with DRIE process to etch away excess silicon in the top and bottom substrates; (**c**) etching SiO_2_ with HF; (**d**) with oxidizing process to form a SiO_2_ layer; (**e**) with electron beam evaporation process to grow Cu layer on the surface; (**f**) with photolithography technology to etch away the Cu and SiO_2_ on the cantilever; (**g**) with doping technology to change the resistivity of the cantilever surface.

## Data Availability

The original contributions presented in the study are included in the article, further inquiries can be directed to the corresponding author.

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
