# Peer review of "Design and Simulation of High-Temperature Micro-Hotplate for Synthesis of Graphene Using uCVD Method"

_micromachines, 2024, doi:10.3390/mi15040445_

Round 1

Reviewer 1 Report

Comments and Suggestions for Authors

Although CVD is already a mature technology, uCVD based on SOI is still a novel and promising research topic. This article describes a fast, convenient, miniaturized, and low-cost improved uCVD graphene growth system suitable for scientific research and experimental needs. The core component of the system is a micro-hotplate. By improving the micro-hotplate,the preparation area of graphene can be increased, and the convenience, reliability, and controllability of the system can be improved. The paper is innovative to a certain extent, but the following aspects still need further improvement:

1.      The quality of English needs to be improved. We strongly suggest that you obtain assistance from a colleague who is well-versed in English or whose native language is English.

2.      On page 4, the word "respectively" should be added after the statement "Where, ci and mi are the specific heat capacity and mass of each layer material, respectively...".

3.      In Figure 6 on page 6, the color scale numbers are blurry and a clearer picture format should be used.

4.      On page 7, ‘.’ should be added at the end of the Funding content.

5.      The reference format should refer to the journal template. For example, in references 1 and 2, the author’s name should not be in all capital letters. The first letter of the journal name in reference 15 should be capitalized, that is, "nature" should be "Nature";

In summary, my review opinion for this paper is that it will be accepted after minor revision.

Comments on the Quality of English Language

Minor editing of English language required

Author Response

Thanks for encouraging us to improve our work.

============

Reviewer 1 #1
============
The quality of English needs to be improved. We strongly suggest that you obtain assistance from a colleague who is well versed in English or whose native
language is English.
Response: Thanks. We have improved the writing quality of the paper.
============

Reviewer 1 #2
============

On page 4, the word "respectively" should be added after the statement "Where, ci and mi are the specific heat capacity and mass of each layer material, respectively...".
Response: Thanks. We have corrected the sentence.
============

Reviewer 1 #3
============

In Figure 6 on page 6, the color scale numbers are blurry and a clearer picture format should be used.
Response: Thanks. We have used clearer images in the revised manuscript
============

Reviewer 1 #4
============

On page 7 7, ‘..’ should be added at the end of the Funding content.
Response: Thanks. We've made the correction.
============

Reviewer 1 #5
============

The reference format should refer to the journal template. For example, in references 1 and 2, the author’s name should not be in all capital letters. The first letter of the journal name in reference 15 should be capitalized, that is, "nature" should be " Nature";
Response: Thanks. We've made the correction.

Reviewer 2 Report

Comments and Suggestions for Authors

The Authors reported on “Design and Simulation of High-Temperature Micro-hotplate for

Synthesis of Graphene Using uCVD Method”, the text is clear and well written, the results are well explained, however still there are some issues that the authors need to address:

1.       The authors need to change the introduction, because most of the work is about the modeling and simulation of the silicon oxide micro-hotplate, not about the graphene and its synthesis.

2.       The authors studied the effect of the micro-hotplate dimension, the silicon resistivity etc. what about the effect of the gas flow (rate and direction) on the heating performances?

3.       What about the effect of the main box volume on the graphene synthesis??

4.       The authors need to change the figure 5, it is not clear.

Author Response

Response to Reviewer 2:

============

Reviewer 2 #1

============

The authors need to change the introduction, because most of the work is about the modeling and simulation of the silicon oxide micro-hotplate, not about the graphene and its synthesis.

Response: Thanks. The main task described in this article is to design micro hotplates that can be used in the preparation system of uCVD graphene. Therefore, the introduction chapter provides more information on micro-hotplate. The synthesis process and product performance of graphene will be the focus of our next work.

============

Reviewer 2 #2

============

The authors studied the effect of the micro-hotplate dimension, the silicon resistivity etc. what about the effect of the gas flow (rate and direction) on the heating performances?

Response:  Thanks. We have added section 4.3 to discuss the effect of gas flow rate on the heating performance of micro hotplate.

============

Reviewer 2 #3

============

What about the effect of the main box volume on the graphene synthesis??

Response:  Thanks. Similar to review comment 1, this content is the focus of our
next paper.

============

Reviewer 2 #4

============

The authors need to change the figure 5, it is not clear.

Response:  Thanks. We have used clearer images in the revised manuscript.

Round 2

Reviewer 2 Report

Comments and Suggestions for Authors

The introduction needs to be improved, authors should add a paragraph regarding the modeling and smulation, otherwise it seems to the reader that the paper will present results regarding the real application of the microCVD synthesis of graphene. 

Author Response

Thank you for pointing this out. We agree with this comment. Therefore, according to the review comments, we have added a paragraph discussing models and simulations at the end of the introduction.